# Communication of preclinical emergency teams in critical situations: A nationwide study

**Matthias Zimmer** [1]*, **Daria Magdalena Czarniecki**[2], **Stephan Sahm**[1,3]

**1** Department of Internal Medicine I, Ketteler Hospital, Hesse, Germany, **2** Department of Psychiatry, Psychotherapy and Psychosomatics, Clinic Hohe Mark, Oberursel, Hesse, Germany, **3** Dr. Senckenberg Institute for History and Ethics of Medicine, Goethe-University Frankfurt, Frankfurt, Hesse, Germany

* zimmer.m@ketteler-krankenhaus.de

## Abstract

### Background

The emergency medical service as a high-risk workplace is a danger to patient safety. A main factor for patient safety, but also at the same time a main factor for patient harm, is team communication. Team communication is multidimensional and occurs before, during, and after the patient's treatment.

### Methods

In an online based, anonymous and single-blinded study, medical and non-medical employees in the emergency medical services were asked about team communication, and communication errors.

### Results

Seven hundred and fourteen medical and non-medical rescue workers from all over Germany took part. Among them, 72.0% had harmed at least one patient during their work. With imprecise communication, 81.7% rarely asked for clarification. Also, 66.3% saw leadership behavior as the cause of poor communication; 46.0% could not talk to their superiors about errors. Of note, 96.3% would like joint training of medical and non-medical employees in communication.

### Conclusion

Deficits in team communication occur frequently in the rescue service. There is a clear need for uniform training in team and communication skills in all professions.

## Introduction

Acute and critically ill patients require fast and precise life-saving treatment. The German Emergency Medical Service (EMS) handles more than eleven million missions a year. The

**Data Availability Statement:** All relevant data are within the manuscript and its Supporting information files.

**Funding:** The authors received no specific funding for this work.

**Competing interests:** The authors have declared that no competing interests exist.

function of the EMS teams is to stabilize patients and make them ready for transport. They then transport the patient with monitoring to a hospital for further treatment. The preclinical phase is characterized by a lack of clarity, information gaps, and limited scope for action. This causes mistakes, failures, and errors in patient care [1]. Team communication is essential for good cooperation for the benefit of the patient [2]. At the same time, dysfunctional communication can harm the patient [3]. Currently, there is only a small amount of data available on communication in the EMS.

During their work, the members of the EMS have to overcome numerous challenges such as those regarding confusing locations, weather influences, limited diagnostics, information deficits, divergent statements by patients and eyewitnesses, and cooperation with other institutions. In addition, there are dangers to one's own health, e.g. from unsafe accident sites, aggressive patients, and physical influences. The fulfilment of orders is managed under time pressure by so-called ad hoc teams [4] whose members belong to different professions. The team members only come together for specific occasions. The crews of the ambulances can be newly combined daily. In addition, different ambulance teams can work together on each mission. Depending on the leadership style, asymmetric relationships may arise between medical and non-medical staff as well as among non-medical staff. The EMS meets the criteria of a high-risk workplace as may be shown by the confrontation with complex situations, temporary lack of resources, and intricate interactions of teams that have to cooperate [4–10]. Regarding the mechanisms that trigger errors, the rescue service is similar to the aviation or chemical industry [11, 12].

Inadequate communication and teamwork are a threat to patient safety. They result in unsafe acts from which errors can develop [12]. Communication is omnipresent. Watzlawick's first axiom is "one cannot not communicate" [13]. Each team member consciously or unconsciously interacts with other team members. Any communication deficit may have an impact on patient safety through their communication behavior.

A study conducted in the Netherlands showed the need for a structured approach to improve communication skills of emergency teams [14]. Williams et al. reported how difficult it is in the Australian EMS to talk about the behavior of a colleague [15]. Research about communication deficits within EMS teams is missing. The aim of the present study was to investigate perceptions of communication deficits and teamwork in the EMS as a cause of harm to patients. In addition, starting points to improve communications and error management were to be identified.

## Materials and methods

In an online based study emergency health care workers were interviewed. The study was descriptively planned, voluntary, anonymous, single-blinded, and without monetary compensation for the participants. We sent non-personalized invitations by e-mail and letters to 1000 German EMS stations and to regional medical directors of EMS who wanted to support the study, medical associations, and promoted the study in an EMS professional journal. The heads of the EMS stations were called upon to make their staff aware of the study. We used the program www.surveymonkey.de to collect responses to the questionnaire. Data were collected online from 01 August 2016 to 20 April 2017. As an indication of the regional distribution of the participants, the postcode of the place of residence was recorded.

In terms of social empirical research, we used a standardized questionnaire survey to achieve a high degree of objectivity in implementation. We wanted to record the purely subjective experiences and opinions of the target group. The questionnaire consisted of single and multiple-choice questions as well as open questions. They were based on 17 hypotheses

regarding communication and patient safety during emergency missions. The individual items of the questionnaire were selected from a previously created universe of items. The universe of items was created based on a thorough literature search in the main areas of communication and risk management. The test structure was subjected to an expert rating. We did not perform a classical hypothesis-based case number calculation because we planned a descriptive, non-interventional study.

As members of the Goethe-University of Frankfurt we were consulted by the Institute for Biostatistics and Mathematical Modelling at the Centre for Health Sciences, Goethe-University Frankfurt. The ethics commission of the State Medical Association of Hesse saw no need for a formal ethical review as the data were being collected anonymously (decision reference number FF67/2016). The study was conducted in accordance with the tenets of the Declaration of Helsinki. The participants were invited in writing via all German EMS stations, the medical directors of EMS in Germany.

### Inclusion criteria

1. Paramedics (PMs) with two years training according to the law on the profession of emergency paramedic.

2. Emergency paramedics (EPMs) with three years training according to the respective medical associations of the German federal states.

3. Emergency physicians (EPs).

4. Staff of a German EMS.

5. Voluntary and unpaid participation in the study.

6. Consent to the privacy policy.

### Exclusion criteria

1. Failure to meet one or more inclusion criteria.

### Data analysis

Due to its descriptive nature, we did not perform classical hypothesis-based case number calculations. The questionnaire consisted of 53 questions which, in addition to demographic information, asked about attitudes with respect to communicative behavior, experiences of errors in patient care and resulting consequences, and associations between communication deficits and maltreatment of patients during care. All statistical analyses were performed using BiAS version 11.06 for Windows (epsilon-Verlag, Frankfurt, Germany).

### Results

Of the 722 sample data received, 714 met the inclusion criteria. Participants (female 17.7%, male 82.3%) had an average age of 35.9±10.5 years and reported an average of 12.5±9.4 years of work experience. PMs (53.5%), EPMs (18.4%), and EPs (28.1%) participated from all over Germany Table 1. The geographical distribution of participants within Germany was homogeneous.

**Table 1. Participants.**

|  | EP | EPM | PM |
|---|---|---|---|
| share of participants | 28.1% | 18.4% | 53.5% |
| male | 76.6% | 91.6% | 82.2% |
| female | 23.4% | 8.4% | 17.8% |
| age, years | 43.0±9.9 | 35.4±9.3 | 32.1±9.5 |
| professional experience, years | 15.2±8.8 | 14.1±8.3 | 10.2±8.6 |

25.1% of the participants expressed a very high level of interest in the topic of communication and 53.3% a high level.

## Patient harm

72.0% the participants stated that they had harmed a patient through their work. In 5.6% of cases, this harm led to disability or death.

## Communication errors

We asked participants about their self-perception of their own communication as receivers and senders of information. Thereby self-reported communication behavior was heterogeneous Table 2.

Communication deficits during patient care can be very different in kind. We operationalized interesting aspects with a focus on the technique of closed-loop communication (i.e. message given, repeated, confirmed) Table 3. In all, 89.7% indicated that the combination of certain team members leads to communication errors more frequently.

Working in EMS can be stressful. Stress as a physiological reaction of the body can lead to a change in perception and behavior. When the participants have to work in stressful situations, they reported a change in their communication Table 4.

When asked about general reasons for poor communication, the participants responded with character traits of colleagues (85.7%), leadership behavior (66.3%), work organization (51.3%) and character traits of one's own person (34.6%). Participants justified their own poor

**Table 2. Characteristics of communication skills as self-estimated by emergency staff.**

| Statements | All the time | Often | Rarely | Never | I do not know |
|---|---|---|---|---|---|
| "During deployment, I communicate precisely and effectively." | 20.9% | 73.1% | 5.8% | 0.3% | - |
| "In patient care, I forget what my colleague said to me." | 0.1% | 10.8% | 74.3% | 14.9% | - |
| "In patient care, I hear from colleagues not appreciative statements about my person." | 0.9% | 4.51% | 34.4% | 60.3% | - |
| "I do not understand what colleagues want from me." | 0.3% | 7.5% | 81.7% | 5.9% | 4.6% |
| "My colleagues do not understand what I want from them." | 0.0% | 4.3% | 77.5% | 12.5% | 5.2% |

**Table 3. Components of a closed-loop communication.**

| Statements | All the time | Often | Rarely | Never |
|---|---|---|---|---|
| "In patient care, I speak to my colleague by name when I pass a task on to him or her." | 30.2% | 49.7% | 16.9% | 3.2% |
| "In patient care, I always know I'm meant when I'm given tasks." | 25.5% | 65.9% | 8.1% | 0.6% |
| "In patient care, I repeat instructions I receive." | 13.5% | 29.8% | 40.9% | 15.8% |
| "In patient care, I report when I have completed a delegated task." | 29.0% | 44.5% | 21.4% | 5.1% |

**Table 4. Communication in stressful situations.**

| Statements | All the time | Often | Rarely | Never |
|---|---|---|---|---|
| "When I'm very stressed, I confuse things." | 0.6% | 5.4% | 63.4% | 30.7% |
| "If I'm very stressed, then I'll interrogate myself." | 0.7% | 8.9% | 60.7% | 29.8% |
| "When I'm very stressed, I don't express myself accurately." | 1.1% | 18.5% | 57.7% | 22.6% |
| "When I'm very stressed, I communicate less and less." | 3.8% | 28.0% | 47.8% | 20.4% |
| "When I'm very stressed, I'll adopt the wrong tone." | 0.9% | 8.5% | 52.4% | 38.3% |

**Table 5. Effects of poor communication when it harms patient.**

| Statements | |
|---|---|
| Negative feelings due to lack of professionalism | 85.1% |
| Shame due to communication deficits | 37.6% |
| Fear of sanctions | 31.3% |
| No negative feelings, because errors are part of everyday | 11.4% |
| Open conversation about communication errors with colleagues | 90.1% |
| Open conversation about communication errors with superiors | 46.0% |

**Table 6. Future training in communication.**

| Statements | |
|---|---|
| Participants could learn something from their colleagues | 89.4% |
| Wish for future theoretical training | 65.9% |
| Wish for future practical communication skill training | 56.9% |
| Wish for supervisions | 43.7% |
| Wish that the topic of communication competence should be integrated into professional training | 76.6% |
| Wish for joint training of physicians and non-physicians in communication | 96.3% |

communication with time pressure (35.1%) and multiplicity number of tasks (58.4%). 14.2% did not want to appear unfocused and therefore tend to poor communication. We believe that current deficits in communication during patient care will have an impact on communication behaviors in future EMS missions. The participants reported about experienced and feared effects of harmful communication Table 5.

As an additional indicator of the meaningfulness of team communication, we see the participants' interest in the wish to improve their own skills of communication Table 6. In all, 43.2% stated that they fully agreed and 45.8% rather agreed that general communication standards and treatment guidelines should be used in EMS missions.

Examination for associations between age, years of experience, and professions revealed no clear links.

## Discussion

With respect to age, sex, and professional training as EMS worker, interviewees are representative for Germany [16, 17]. The number of evaluable questionnaires is noteworthy in view of the informal invitation to participate. This may reflect the high level of interest in the topic of communication. For the first time the present study provides an insight into the perception and experience of communicative behavior and its deficits in the EMS.

## Patient harm

The number of emergency staff indicating having caused harm to patients by communication deficits is high. For the first time we are able to quantify the degree because other studies had only researched selected cases [3]. The extent of patient harm is sometimes considerable (disability or death). The experience of mistakes seems to be an everyday occurrence that is little known to the general public (population, EMS staff) outside of small expert circles. The evaluation of the German Critical Incident Reporting System for Emergency Medicine [3] identified a deficit in team communication as the trigger for 27% of cases of patient harm. The currently reported error frequency is not tolerable, and the self-assessment probably underestimates the true frequency of error.

## Communication errors

The majority of the interviewees attested themselves a rather good communication behavior. The low level of reported misunderstandings also fits in with this conclusion. However, this is subjective self-perception, which can be distorted. At the same time, the severity of the communication deficit does not correlate with its impact. According to J. Reason's Swiss Cheese model (cumulative act effect) [1], even minor misunderstandings can have serious consequences.

Due to the heterogeneous results, we suspect that the concept of closed-loop communication [18, 19] is not comprehensively known in German EMS. Probably the participants use components of this concept rather unconsciously. But the conscious use of closed-loop communication could be effective in reducing communication deficits [19]. And although the team members were rarely addressed by name, they seem to know more often that they were being addressed. This may be due to the fact that the majority of EMS deployments only occur in teams of two. Or there is non-verbal communication. The fact that the participants attested themselves a good communication behaviour and at the same time they did not master closed-loop communication completely, points to clear knowledge gaps.

We suggest the integration of closed-loop communication into the EMS training. EMS teams should use this communication tool in any emergency, no matter how uncomplicated, to reduce misunderstandings and their consequences.

The participants lack strategies to maintain effective communication in stressful situations. This creates an alarming threat to patient safety. Deficits in vocational training are most likely. Again, there is a discrepancy between self-assessment and requirements for good communication in high-risk workplaces.

The causes of poor communication are more likely to be seen in other team members than in themselves. We suggest that behind this result is a lack of ability to introspect and a lack of understanding of the mechanisms of team communication. More experienced employees see considerable causes for poor communication in the leadership behavior of their superiors and the organization of work. The organizational structure of EMS in Germany demands formation of ad hoc teams. This seems to be a key element in causing errors as the teams are not well trained for this kind of cooperation. The different professional groups are neither interlocked in training nor can they practice collegial cooperation outside work assignments. In addition, there is an asymmetrical relationship between the physician and non-physician team members due to differences in professional training and role perception. This asymmetry has not yet been balanced out or addressed constructively.

The consequences of poor communication are significant for the participants. Fear of sanctions, shame and the loss of reputation are major outcome risks. They do not talk to their own superiors. This makes it difficult for superiors to uncover systematic problems and to improve

patient safety. Davidoff reported 17 years ago [20] that medical professions still have an inadequate way of dealing with guilt and shame. The maintenance of the utopian zero-defect principle counteracts the constructive handling of mistakes [21].

Communication as an integral part of sufficient teamwork is not adequately recognized. However, the participants expressed a broad interest in interdisciplinary education and training in communication. In addition, they wished for establishing binding communication standards and treatment.

Team communication must be perceived as an important source of error that can be addressed. It is surprising that pilots receive detailed instruction on decision-making and the influence of the human factor during their training, while EPs do not learn anything about team communication either during their university studies or during their additional training in emergency medicine [22]. The advanced training curriculum for additional qualification in emergency medicine must be intensified urgently with regard to cooperation within a team and communication skills.

The same applies to medical assistance staff. During training of EPM they will receive 105 hours of teaching in the area of team resource management (theory, exercises, patient care simulations) [23]. In addition, there are further theoretical lessons on specified communication fields. It must be noted that the concrete learning contents are not predetermined. Moreover, most EPMs are former PMs who had to pass only a medical knowledge examination without communication training. Regular annual EMS refresher courses that are obligatory should also include communication training in the future. As well, guidelines for structured communication in emergency care have to established and should be adopted across borders of federal states.

However, our data cannot clarify how German conditions affect communication and teamwork. The ability to communicate in a team must become an indispensable prerequisite for taking up a career in the EMS. At the same time, a change in the approach to errors and the human factor involved should occur. An uninhibited handling of mistakes is needed, which allows constructive debriefings and supervision. The rescue service must learn from its mistakes and continually improve patient safety.

## Limitations

In interpreting the data presented, it should be borne in mind that more motivated and interested employees of the rescue service might have responded. Employees with less interest in the topic or their profession or reduced communication skills may have declined to participate. The study is not an objective observation but provides information about subjective perceptions. Emergency medical technicians, who are widespread in the rescue service for cost reasons and receive only a short professional training, were not interviewed.

## Supporting information

**S1 Dataset. Dataset of communication of preclinical emergency teams in critical situations: A nationwide study.**
(XLSX)

**S1 File.**
(PDF)

**S2 File.**
(PDF)

## Acknowledgments

We thank the regional medical directors of EMS, the Institute for Biostatistics and Mathematical Modelling at the Centre for Health Sciences, Goethe-University Frankfurt for their help and all participants for their participation in our study.

## Author Contributions

**Conceptualization:** Matthias Zimmer, Daria Magdalena Czarniecki, Stephan Sahm.

**Data curation:** Matthias Zimmer, Daria Magdalena Czarniecki.

**Formal analysis:** Daria Magdalena Czarniecki.

**Investigation:** Matthias Zimmer, Daria Magdalena Czarniecki.

**Methodology:** Matthias Zimmer, Daria Magdalena Czarniecki.

**Project administration:** Matthias Zimmer.

**Software:** Matthias Zimmer.

**Supervision:** Stephan Sahm.

**Validation:** Matthias Zimmer.

**Visualization:** Matthias Zimmer, Daria Magdalena Czarniecki.

**Writing – original draft:** Matthias Zimmer, Stephan Sahm.

**Writing – review & editing:** Matthias Zimmer, Stephan Sahm.

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
