## [Decision Letter · Decision Letter 0]

2 Nov 2020

PONE-D-20-17178

Communication of preclinical emergency teams in critical situations: A nationalwide study

PLOS ONE

Dear Dr. Zimmer,

Thank you for submitting your manuscript to PLOS ONE. After careful consideration, we feel that it has merit but does not fully meet PLOS ONE’s publication criteria as it currently stands. Therefore, we invite you to submit a revised version of the manuscript that addresses the points raised during the review process.

We look forward to receiving your revised manuscript.

Kind regards,

Bernadette Watson, Ph.D.

Academic Editor

PLOS ONE

Additional Editor Comments:

I have read through the Reviewer's comments and completely agree with the comments. However, I do feel that the paper needs substantial editing before it can is publishable.

Your method section is incomplete. You superficially describe the participants. You say in the discussion that they are representative but although you provide percentages with respect to gender, age and work experience in the reporting of your participants, I think you need to provide more information. Specifically, what is the breakdown of each profession and years of experience. It would be intriguing to know if the responses in some way differ between professions and experience. Again, this need only be descriptive but would bring your data to life. If there really are no differences then say so. Again, given that you are not conducting inferential statistics, why do you run non-parametric statistical analyses. The fact that you have such a large number of participants means you could have run inferential statistics. What was the reason for not doing so? It is fine to be descriptive but the overwhelming numbers in the tables is at odds with the emphasis that it is a descriptive paper. From what I can see the statistical analysis is redundant. I think you could detail the items used in the survey and state where they were sourced more fully. Why did you choose these questions? I assume they relate to the table headings but you do not introduce them in this way. What was your overarching research question under which these 53 questions sit?

I also want to know how the survey was organised. Did the participants return in a prepaid envelope? What was the response rate? These details are important.

One of the reasons that I believe you made mistakes with respect to the wrong information and table numbers is because you just have too many tables and they are presented unclearly. You need to reformat your tables so that they are not so cluttered. One way to achieve this would be to remove the confidence intervals. You are reporting descriptive statistics so simply report the percentages and numbers. This would make your tables more readable and uncluttered. Why are there no tables for the final three items discussed in the results? It seems strange given that you follow this format until that point.

I wonder if you might consider combining results and discussion sections. After each table you could discuss the implications of what the numbers mean. This would improve comprehension of the paper and make for a more interesting and varied format. You can then conclude with a section on what it all means and what needs to be done.

In your discussion you talk about more experienced members differing in terms of leadership etc but there is no such information in the tables themselves.

In summary I am suggesting your reformat the paper and in so doing bring the data to life. I know this requires a great deal of work but if you are prepared to do this, it will increase the value of the paper immensely.

Journal Requirements:

2. During your revisions, please note that a simple title correction is required: "Communication of preclinical emergency teams in critical situations: A nationwide study". Please ensure this is updated in the manuscript file and the online submission information.

Please amend either the title on the online submission form (via Edit Submission) or the title in the manuscript so that they are identical.

"We thank the medical directors of the rescue service, the Institute for Biostatistics and

 Mathematical Modelling at the Centre for Health Sciences, Goethe-University Frankfurt for

 their support and all participants for their participation in our study."

"The authors received no specific funding for this work."

Reviewers' comments:

Reviewer's Responses to Questions

**Comments to the Author**

1. Is the manuscript technically sound, and do the data support the conclusions?

Reviewer #1: Yes

2. Has the statistical analysis been performed appropriately and rigorously? 

Reviewer #1: Yes

3. Have the authors made all data underlying the findings in their manuscript fully available?

Reviewer #1: No

4. Is the manuscript presented in an intelligible fashion and written in standard English?

Reviewer #1: No

5. Review Comments to the Author

Reviewer #1: Thank you for doing this research and writing up the report for publication. It reads very well and is a very important underreported issue that requires attention. The manuscript is likely to appeal to communication scholars across medical. EMS and allied health, but also to social scientists interested in intergroup communication in healthcare. Some word and phrase choices confused me and made me question your statements and wonder if you have made some errors in reporting (a paragraph in particular L 148-167) because the table numbers were missing or wrongly matched to statements. Otherwise, I make a few suggested edits for clarity mostly and some for emphasis to enhance message. My edits are noted next and follow line numbers in manuscript (line number 'L'). Suggestions: L62 add reference? L66 edit to 'have an impact on patient safety' L99-106 copyedits needed capitalisation and periods L115 edit to 'patient care' L143 start sentence with 'Participants reported ..' L148-167 Review errors identified in text: Statements, table numbers and statistics L207 Consider '... communication standards and guidelines' (for clarity and emphasis) L209 unclear - pls review L215,216 consider edit to read '.. an insight into perceptions and experience of communicative behaviour and ..' L249 'completing incomplete' - confusing; consider rewording L251 reconsider word choice 'major aftermaths' - suggest change to 'major outcome risks' for example L259 suggest edit to 'However, the participants expressed a broad..' L261 for clarity suggest edit to 'binding communication standards and ..' if appropriate and true L270 review for edits word choice and order L276 suggest edit to 'The ability to communicate ..' for emphasis and clarity L280 replace 'as well' with 'also' L282 insert comma 'At the same time, a ..' L290 suggest edit to '... may have declined to participate ..'. I believe these edits will enhance the readability and flow and mitigate risk of confusion. Good work and good luck.

6. PLOS authors have the option to publish the peer review history of their article (what does this mean?). If published, this will include your full peer review and any attached files.

Reviewer #1: No

---

## [Author Response · Author response to Decision Letter 0]

15 Feb 2021

Dear Sir or Madam,

Thank you for giving us the opportunity to revise our manuscript. We have carefully considered your suggestions and have made some extensive modifications.

We have specified the description of the participants in terms of age, gender and work experience. The corresponding data are now presented in Table 1.

Statistical analyses were performed under the advice of the Institute for Biostatistics and Mathematical Modelling at the Centre for Health Sciences, Goethe-University Frankfurt. We cannot retrospectively substantiate the rationale for omitting an inferential statistic. However, we agree with your assessment and have removed the references to statistical analysis. We have restricted ourselves to a pure description of the results.

We have taken up reviewers´ suggestions and commented on the selection of items.

As had been proposed we have described in detail the procedure of invitation to take part in the study and the process of data collection. That my help readers´ perception of the study. Participants were not contacted directly because we did not know them personally. Rather, we sent general invitations to participate to the Emergency Medical Services (EMS) stations. In parallel, we asked the EMS regional medical directors for additional forwarding of the invitations to the subordinate EMS stations. Not all directors supported us. Participation was via a link to the world wide web and involved only a small investment of time for participants. There was no compensation for costs and no other contributions.

We cannot give detailed figures about response rate because it is not known how many employees in the EMS received the invitation.

At your suggestion, we have revised the content and structure of the results and discussion sections. The order of the discussion is now parallel to the presentation of the results. We prefer to present the results and the discussion separately, since our hoped-for target group is from the medical field and is strongly accustomed to this form of presentation.

Unfortunately, we did not adequately describe the context in the acknowledgement. The regional medical directors of EMS did not fund us, but authorized letters of invitation in their counties and forwarded them to the subordinate EMS stations. The Institute for Biostatistics and Mathematical Modelling at the Centre for Health Sciences, Goethe-University Frankfurt helped collegially as we are faculty members of Goethe-University Frankfurt and this is the mission of the Institute. There are no financial or other dependencies between us and the institute. We kindly ask you to reassess the facts.

Reviewer 1

We thank you for the detailed review comments and have carefully edited all points. At the same time, we have redesigned result and discussion section, respectively, to make them easier to understand. With the restriction to a purely descriptive presentation of the results and after the redesign of the results, all existing data are available to you.

We hope to have responded intensively to all your suggestions and hope for an acceptance of our article in PLOS ONE. 

With best regards

Matthias Zimmer

Daria M. Czarniecki

Stefan Sahm

---

## [Decision Letter · Decision Letter 1]

22 Mar 2021

PONE-D-20-17178R1

Communication of preclinical emergency teams in critical situations: A nationwide study

PLOS ONE

Dear Dr. Zimmer,

Thank you for submitting your manuscript to PLOS ONE. After careful consideration, we feel that it has merit but does not fully meet PLOS ONE’s publication criteria as it currently stands. Therefore, we invite you to submit a revised version of the manuscript that addresses the points raised during the review process.

We look forward to receiving your revised manuscript.

Kind regards,

Bernadette Watson, Ph.D.

Academic Editor

PLOS ONE

Journal Requirements:

Additional Editor Comments (if provided):

The reviewer has noted the improvements to this paper and its importance. I concur with the reviewer's opinions. I would ask you to take note of the small revisions recommended. They should not take long.

Regards

Reviewers' comments:

Reviewer's Responses to Questions

**Comments to the Author**

1. If the authors have adequately addressed your comments raised in a previous round of review and you feel that this manuscript is now acceptable for publication, you may indicate that here to bypass the “Comments to the Author” section, enter your conflict of interest statement in the “Confidential to Editor” section, and submit your "Accept" recommendation.

Reviewer #1: (No Response)

2. Is the manuscript technically sound, and do the data support the conclusions?

Reviewer #1: Yes

3. Has the statistical analysis been performed appropriately and rigorously? 

Reviewer #1: Yes

4. Have the authors made all data underlying the findings in their manuscript fully available?

Reviewer #1: Yes

5. Is the manuscript presented in an intelligible fashion and written in standard English?

Reviewer #1: Yes

6. Review Comments to the Author

Reviewer #1: Many thanks for considering and integrating suggestions offered in my initial review of the manuscript into the current version. I thank you for your letter, and I accept your explanations with gratitude as these have helped my understanding of the purpose, execution and results of the research. The current manuscript is informative, non-biased, reads well, is clear, and delivers an important message for all stakeholders - EMS staff, health communication educators, health leaders, patients and the friends and families that support them.

I note there are some very minor edits remaining (tense consistency and punctuation). Please read and edit or reconsider the following (indicated by line number). Square brackets indicate suggestions and questions about word use.

27 single-blinded study

34 (Delete 'Again') Of note, 96.3%

46 At the same time, (add comma)

55 ...newly combined daily. In addition, ...

71 ...investigate perceptions of communication deficits ...

72 ...management were to be identified.

76 single blinded

79 (Delete 'made a call')... promoted the study in an EMS ...

80 ...heads of EMS stations were called upon to make...

81 ...collect responses to the questionnaire.

128 ??? What do you mean by participated 'equally' as you give different percentages!

143 (149 & 154) Tables. Use the word 'rarely' not 'rare' (it's an adverb not adjective). Add periods to end of statements in tables.

146 ...closed-loop communication (i.e. message given, repeated, confirmed). [I think it's better to provide a little information to the reader about what closed-loop communication is in case they don't know. So please add this in parentheses as shown.]

149, 154 ['rarely' and periods as above stated]

158 Participants justified...

160 did not want...

172 workers, interviewees...

182 ??? [The first, and third sentences seem to be contradicted by the fourth. Review and rewrite for clarity. For example make clear that the percentage refers to units in hospitals (?) but we didn't know about EMS experience because it is not yet published - this is my guess at least.]

194 ...conclusion. However, ...

200, 202, 207 closed-loop

219 ...introspect

230 ...risks

253 As well, ...

257 ??? [More or less than what? This is unclear to me as it sounds like a comparison but I am guessing.]

259 ...and the human factors involved should occur.

266 [This is a suggestion only: some of the data and your statements refer to shame (e.g. line 230), and I wonder if shame may have been a barrier to participation for those with an experience of communication failure (e.g. line 136; 72% experienced communication failure). Not to conflate the issue, but shame is a predictor of poor mental health and suicide ideation. As such, if shame was a barrier to participation, there are vulnerable service providers keeping quiet about communication failures and who may be at risk.]

Thank you for considering these final edits (there may be others - please review carefully!)

Congratulations on your research success, and thank you for taking the time to ensure the scientific community is informed of this important factor in EMS delivery. I hope the publication receives the attention it deserves, and supports EMS staff, their own selfcare, and patients receiving emergency services into the future.

Kindest regards, Lori.

7. PLOS authors have the option to publish the peer review history of their article (what does this mean?). If published, this will include your full peer review and any attached files.

Reviewer #1: **Yes: **Lori Ellen Leach

---

## [Author Response · Author response to Decision Letter 1]

9 Apr 2021

Dear Sir or Madam,

Thank you for giving us the opportunity to enhance our manuscript. We have taken up all reviewers´ suggestions and have made precise corrections. In particular, we have clarified two passages in the discussion.

We thank you for the productive cooperation and hope for an acceptance of our article in PLOS ONE. 

With best regards

Matthias Zimmer

Daria M. Czarniecki

Stefan Sahm

---

## [Editor Report · Decision Letter 2]

19 Apr 2021

Communication of preclinical emergency teams in critical situations: A nationwide study

PONE-D-20-17178R2

Dear Dr. Zimmer,

We’re pleased to inform you that your manuscript has been judged scientifically suitable for publication and will be formally accepted for publication once it meets all outstanding technical requirements.

Kind regards,

Bernadette Watson, Ph.D.

Academic Editor

PLOS ONE

Additional Editor Comments (optional):

Thank you for your attention to these revisions. The paper now reads well and is a valuable contribution to the area.
---

## [Editor Report · Acceptance letter]

23 Apr 2021

PONE-D-20-17178R2 

Communication of preclinical emergency teams in critical situations: A nationwide study 

Dear Dr. Zimmer:

I'm pleased to inform you that your manuscript has been deemed suitable for publication in PLOS ONE. Congratulations! Your manuscript is now with our production department. 

Kind regards, 

on behalf of

Dr. Bernadette Watson 

Academic Editor

PLOS ONE